# Comfortable Seatbelts for Pregnant Women with Twins in Japan: A Cross-Sectional Observational Study on Seatbelt Usage

**DOI:** 10.3390/healthcare12161590

**Published:** 2024-08-09

**Authors:** Sachi Tsuchikawa, Yui Miyajima, Yumiko Tateoka, Masahito Hitosugi

**Affiliations:** 1Department of Nursing, Shiga University of Medical Science, Otsu 520-2192, Japan; ns251953@g.shiga-med.ac.jp (Y.M.); ytateoka@belle.shiga-med.ac.jp (Y.T.); 2Department of Legal Medicine, Shiga University of Medical Science, Otsu 520-2192, Japan; hitosugi@belle.shiga-med.ac.jp

**Keywords:** twins, pregnancy, vehicles, driving, seatbelt, health education

## Abstract

Optimal seatbelt practices for pregnant women with twins at different gestational ages remain uncertain. To offer recommendations for a comfortable seatbelt system, this cross-sectional observational study explored seatbelt usage and driving habits among women with twins across various pregnancy stages through an online survey that explored driving conditions and comfortable seatbelts at different stages of pregnancy. Women who drove daily before their pregnancy with twins decreased their driving frequency as the pregnancy progressed. Correct seatbelt usage was lower and no seatbelt usage was higher among pregnant women with twins than those with singleton pregnancies. They adapted their seatbelt-wearing techniques to minimise pressure on the chest in the first and third trimesters and the abdomen from the second trimester onwards. The comfortable seatbelts were those that could alleviate belt pressure, featuring waist belts to reduce pressure, wider belts to avoid localised pressure, and shoulder belts resembling a backpack type. When wearing a seatbelt, avoiding pressure on the thorax and abdomen is key for pregnant women with twins. This study suggests that the suitability of driving for pregnant women with twins in their last trimester and the reliability of seatbelts designed for such women should be further examined and validated.

## 1. Introduction

The rate of automobile usage among Japanese women is increasing, with >70% of women in their 20s and 30s driving cars or riding motorcycles daily. Furthermore, approximately 90% of women with singleton pregnancies continue to drive post-pregnancy [1]. Notably, approximately 3% of these women report motor vehicle collisions (MVCs) [2,3] if they were drivers before pregnancy, and approximately 10% of women have experienced near-misses or MVCs [4]. Daily driving is a common practice among most Japanese women, including driving during pregnancy. This behaviour increases their risk of traffic-related trauma, which is a leading cause of maternal trauma [5]. Traffic injuries can lead to severe maternal and foetal complications, such as threatened preterm birth, early placental abruption, and uterine rupture [6]. In Japan, the use of three-point seatbelts is mandatory to ensure the safety of both mothers and children, and 97.7% of Japanese women with singleton pregnancies comply with this requirement. However, 13.1% do not wear seatbelts correctly [7] due to insufficient knowledge of how to use them properly and how to avoid putting pressure on the abdomen [7,8]. A previous study on pregnant women carrying twins found that although >90% used seatbelts, only 68.3% wore them correctly. The rate of incorrect usage exceeded 30% in a previous study [9], highlighting a significant gap in road safety practices among pregnant women. This issue underscores the urgent need for enhanced road safety education for pregnant women and their families.

Twin pregnancies are associated with increased risks of obstetric complications. Compared to those with singleton pregnancies, women with twin pregnancies experience more pressure and discomfort from seatbelts earlier in pregnancy due to more pronounced abdominal protrusion and uterine enlargement. As more Japanese women enter the workforce later in life and delay marriage, the demographics of pregnant women are shifting towards older ages [10], and there are increased incidences of infertility treatments [11]. Currently, approximately 1 in every 50 pregnant women carries twins [12]. With the increasing number of twin pregnancies [12], it is crucial for healthcare providers to consider the daily needs of these women. Recent studies have focused on driving and seatbelt use among women with singleton pregnancies; however, there is a lack of research on twin pregnancies in this regard. Therefore, this study aimed to investigate actual seatbelt usage and driving habits among women with twins across various stages of pregnancy and explore the real-world conditions of safe driving for women pregnant with twins. We sought to gather data on their driving habits and seatbelt usage throughout different stages of pregnancy and offer recommendations for designing a comfortable seatbelt system for them.

## 2. Materials and Methods

### 2.1. Design

This cross-sectional observational study was conducted through an online survey distributed via a membership site for survey monitors owned by a contractor (GMO Research & AI, Inc. Tokyo). The contractor, engaged in internet research, sent survey requests to Japan Cloud Panel, a member site owned by the company. Women who consented to participate were asked to complete the survey. With a response rate of 50%, sampling error of 5%, and reliability of 95%, the required sample size was 384. However, based on a previous survey by a contractor with a 95% response rate, the required sample size was 200. There are around 8000 twin pregnancies annually in Japan, and not all are registered with the contracting company. Therefore, to ensure a feasible sample size, we recruited 100 women whose pregnancies continued to the third trimester, including those who had been pregnant with twins within the past three years. In this survey, women who were in the third trimester or had ended their pregnancy were asked to recall their first, second, and third trimesters of pregnancy and respond to the questionnaire regarding these. The questionnaire was created by authors based on previous research on seatbelt use among singleton pregnant women [7,9,13]. This study was approved by the Ethics Committee of the Shiga University of Medical Science (no. RRB21-055-2).

### 2.2. Participants

Women who were either previously or currently pregnant with twins were included in this study. Women who did not drive a car or did not possess a driver’s licence before pregnancy were excluded from this study.

### 2.3. Survey Content

The survey encompassed the following areas:
1.Participant characteristics:
a.Age.b.History of pregnancy and childbirth.c.Possession of a driver’s licence.


Subsequently, the following items were surveyed before pregnancy and at the first, second, and third trimesters:

2.Number of operating days in a week: Categories included ‘almost every day’, ‘a few days a week’, and ‘less than a few days a week’. Reasons for changes in driving frequency were also explored.3.Seatbelt use: Participants indicated their seatbelt usage from seven patterns based on a previous report suggesting the seatbelt use methods among singleton pregnant women (Figure 1):a.Wearing a seatbelt that avoids the distended abdomen.b.The lap belt crosses the most distended part of the abdomen.c.The lap belt crosses the upper abdomen.d.The shoulder belt crosses the upper abdomen.e.The shoulder belt is not worn.f.The lap belt crosses the thighs.g.The shoulder belt crosses the armpit.

The following items were surveyed at the first, second, and third trimesters:

4.Frequency of seatbelt use: Categories included ‘Always wear a seatbelt while driving (did)’, ‘Used to wear a seatbelt but stopped during driving (stopped)’, and ‘Never wear a seatbelt while driving (did not)’.5.Feelings while wearing a seatbelt: Categories included discomfort, pressure, breathlessness, skin itching with the contact of a seatbelt, and increased back pain.6.Devices to fasten seatbelts: Multiple response options were available, including the following:a.No particular method.b.I shifted the seatbelt as close as possible to the base of my legs to avoid abdominal protrusion.c.A piece of cloth was placed under the belt to reduce discomfort at the point of contact with the skin.d.The belt thickness was adjusted by adding fabric to the belt to relieve abdominal pressure.e.Adjusted reclines to prevent prolonged sitting.f.Avoided bulging breasts.g.Used a device.7.Comfortable seatbelts were preferred by pregnant women with twins. Respondents chose the following seatbelts as being comfortable while driving:a.A narrow seatbelt that does not put pressure on the stomach.b.A thicker seatbelt that induces less pressure on the stomach.c.A seatbelt that can be loosened to relieve pressure.d.A belt that secures both thighs instead of the abdomen.e.A seatbelt made of soft material that does not damage the skin.f.A mechanism for extending the buckle to ensure that the wearer does not have to look down diagonally.g.A belt that supports both shoulders, similar to a backpack, instead of a diagonal shoulder belt.8.Sources of information on seatbelt use: Respondents were asked to select multiple answers from the following options:a.Have not received any information.b.Health guidance at medical facilities.c.Acquaintances and family members.d.Maternal and Child Health Handbook.e.Social networking service (SNS).f.‘Traffic instructions‘ distributed during driver’s license renewal.g.Magazines and television.h.Others.

### 2.4. Statistical Analysis

Data were analysed using IBM SPSS Statistics for Windows, version 29 (IBM Corp., Armonk, NY, USA). Statistical significance was set at *p* < 0.05. We performed a Shapiro–Wilk test, which did not confirm normality; consequently, we conducted a nonparametric test. Driving frequency was compared using the Friedman test for the number of days before pregnancy and in each trimester. For multiple comparisons, we employed the Bonferroni method. Seatbelt-wearing methods were analysed using the χ-square test, with the Bonferroni method again used for multiple comparisons. Simple tabulations were performed for other items.

## 3. Results

### 3.1. Participants’ Characteristics

Overall, 115 women responded to the questionnaire, all of whom reported driving a car at least five days per week before pregnancy. Of these respondents, complete data up to the end of pregnancy were available for 82 respondents (71.3%). Among the participants, 68 (82.9%) had a history of twin pregnancy, whereas 14 (17.1%) were currently in the last trimester of their pregnancy. The median age was 36.0 years (interquartile range 28.8–42.3).

### 3.2. Number of Operating Days in a Week

The number of operating days per week before pregnancy and at each trimester was examined and compared. The number of operating days in a week significantly decreased after pregnancy compared to before pregnancy. Notably, in the third trimester, it further decreased significantly compared to that in the first and second trimesters (Figure 2). These findings indicate that women tended to drive less frequently not only before pregnancy but also as the weeks of pregnancy progressed.

Among the pregnant women surveyed, 69.5% reported a decrease in driving frequency, 29.3% reported no change, and 1.2% indicated an increase. The most common reason for reducing driving frequency was ‘because it is a burden to drive’ (30 respondents [52.6%]), followed by ‘because I was afraid of causing a collision’ (25 [43.9%]), ‘because it was difficult to wear a seatbelt‘ (22 [38.6%]), ‘because I had the support of others’ (19 [33.3%]), ‘less purpose for driving’ (10 [17.5%]), and ‘other reasons’ (8 [14.0%]). ‘Other reasons’ included factors such as the perceived risk status of the continuing pregnancy, which led to actions such as hospitalisation until birth, health concerns, medical advice to refrain from driving, or general anxiety about car travel, as indicated by answers including ‘I do not want to get into a collision’.

### 3.3. Seatbelt-Wearing Method

Table 1 shows the number of pregnant women for each method of seatbelt use before pregnancy and during each trimester. The rates were calculated at each stage. The highest proportion of correct seatbelt usage was observed before pregnancy, with 51 women (62.2%) wearing seatbelts appropriately. However, this number decreased as pregnancy progressed (Table 1). Specifically, the proportion of participants not wearing the shoulder belt (‘The shoulder belt is not worn’) increased to approximately 20% by the third trimester, compared to pre-pregnancy and in the first and second trimesters. Additionally, the proportion of women positioning the lap belt across the thighs (‘The lap belt crosses the thighs’) increased significantly from 2 out of 82 (2.4%) pre-pregnancy to 4 out of 24 (16.7%) in the second trimester (*p* < 0.05, both).

### 3.4. Seatbelt-Wearing Frequency

Among the 25 women whose driving frequency either remained constant or increased during pregnancy, the rate of seatbelt usage during each trimester was consistent: 24 participants (96.0%) in the first trimester, 24 (96.0%) in the second trimester, and 21 (84.0%) in the third trimester responded ‘always wearing’. Conversely, one woman (4.0%) in the first and second trimesters and four women (16.0%) in the third trimester did not wear seatbelts while driving. No statistically significant differences were observed in the frequency of seatbelt usage in each period. Reasons for not wearing a seatbelt included ‘I did not have the habit to begin with’ in all trimesters. Additionally, in the third trimester, the reasons included ‘I did not have the habit to begin with’, ‘because I felt pressure in my abdomen’, and ‘other: I was hospitalised and could not ride in a car’.

### 3.5. Feelings of Discomfort While Wearing a Seatbelt

Overall, 18 (75.0%), 10 (41.7%), and 8 (38.1%) respondents in the first, second, and third trimesters, respectively, reported ‘no particular discomfort’, with a statistically significant increase in those experiencing discomfort as the pregnancy progressed (Figure 3). No statistically significant differences were observed for the other items; however, regarding the response ‘I had pressure on the abdomen’, the number increased from 6 in the first trimester (25.0%) to 12 in the second trimester (50.0%). The number of respondents who experienced chest pressure increased from the first to the third trimester. The response ‘breathing was difficult’ remained constant throughout pregnancy. Skin itching from seatbelt contact and back pain increased during the second and third trimesters.

### 3.6. Devices for Seatbelt-Wearing

Despite the limited number of responses, several pregnant women devised methods for wearing seatbelts starting from the first trimester (Figure 4). Notably, those who ‘shifted the seatbelt as close as possible to the base of their legs to avoid abdominal protrusion’ accounted for 66.7% (4 of 6), 28.6% (4 of 14), and 23.1% (3 of 13) of respondents in the first, second, and third trimesters, respectively. Those who ‘adjusted the belt’s thickness, such as adding fabric to the belt to relieve abdominal pressure‘, ‘adjusted the recline to avoid prolonged sitting‘, and ‘avoided bulging breasts’ most likely adjusted in the first trimester. A new adaptation introduced in the second trimester or later involved placing a cloth or other material under the belt to reduce discomfort in the skin and ingestion areas.

### 3.7. Comfortable Seatbelts Preferred by Women Pregnant with Twins

The comfortable seatbelt features identified by the respondents included ‘one that can be loosened to relieve pressure’ (37 respondents [45.1%]), followed by ‘a narrow seatbelt that does not put pressure on the abdomen’ (25 [30.5%]) and ‘a wider seatbelt that does not put pressure on the abdomen’ (23 [28.0%]). Additionally, 20 respondents (24.4%) preferred ‘a belt that supports both shoulders, similar to a backpack, instead of a diagonal shoulder belt’, and 15 (18.3%) favoured ‘a belt that secures both thighs instead of the abdomen’. Ten participants (12.2%) chose ‘a seatbelt made of soft material that does not hurt the skin’ (10 [12.2%]), seven (8.5%) preferred ‘a mechanism for extending the buckle to ensure the wearer does not have to look diagonally down’, and three (3.7%) selected ‘others (something soft anyway)’. Overall, the majority of participants preferred seatbelts that could be loosened to relieve pressure.

### 3.8. Provision of Information on Seatbelt Use

The most common source of information on seatbelt use was, ‘I have not received any information’ (40 [48.8%]) followed by ‘health guidance at the medical facility where I am receiving medical care’, ‘Maternal and Child Health Handbook’, ‘acquaintances and family’, and SNS (Figure 5). Notably, some respondents obtained information from ‘books’ as other sources.

## 4. Discussion

In this study, we examined the driving habits and conditions of seatbelt usage at different stages of pregnancy for women pregnant with twins. According to the World Health Organization, seatbelt use by front-seat occupants reduces the risk of fatal injuries by 45–50% and decreases the risk of death and serious injuries among rear-seat occupants by 25% [14,15]. Therefore, a seatbelt remains the best vehicle safety device to protect passengers from being severely injured in a collision or ejected from the vehicle. Over the past several decades, seatbelt use has been legally required, especially in developed countries. According to the latest WHO Global Status Report on Road Safety 2023 [16], 117 countries have seatbelt usage laws aligned with best practices. Moreover, for pregnant women, wearing a seatbelt has reportedly better foetal outcomes after MVCs compared to not wearing one [17,18]. Therefore, the correct use of seatbelts has been widely advocated for pregnant women drivers and passengers [19].

However, pregnant women often face challenges with seatbelt use, particularly in positioning it correctly, due to physical changes and discomfort [20]. Acar suggested that the size of the chest, abdomen, and hips of a pregnant woman can become significantly larger during pregnancy, surpassing the dimensions of the large 95th percentile for males by a considerable amount [13]. Therefore, substantial numbers of pregnant women are not wearing their seatbelts correctly, and some women may also cease to use their seatbelts during pregnancy. Furthermore, substantial numbers of pregnant women experience discomfort currently by using today’s car seatbelts. To improve these situations, actual conditions of seatbelt use and discomfort due to seatbelts have to be examined and adequate measures should be taken. Previously, for Japanese singleton pregnant women, methods of seatbelt use and the frequency of common pregnancy complaints during driving were examined [4,7]. However, because the physical size and shape of women pregnant with twins are different from those of singleton pregnant women, the current study is essential for promoting the safety of women pregnant with twins. In this study, we collected data from Japanese women pregnant with twins.We have compared some of the results to those obtained from Japanese singleton pregnant women in previous research. Because the size and shape of the body depend on race, we only used data obtained from Japanese women. In the future, similar research should be performed in other countries or regions.

### 4.1. Seatbelt-Wearing Status Based on Pregnancy Period

In this study, the method of ‘Wearing a seatbelt that avoids the distended abdomen’, considered the appropriate way to wear a seatbelt, showed a tendency to decrease towards the end of the pregnancy compared to the pre-pregnancy period. According to a previous report on singleton pregnant women, among drivers who always wore a seatbelt, 86.9% of them did so correctly [7]. In contrast, the present study found that the rate of correct seatbelt use ranged from 42.9% to 54.2% during pregnancy, which is markedly lower. The difference may be attributed to variations in body shape and size. A significant increase was noted in the number of pregnant women who responded that ‘The lap belt crosses the thighs’ in the second trimester and ‘The shoulder belt is not worn’ in the third trimester of pregnancy. These findings, revealed by trimester-based surveys, are a novel aspect of this study. Regarding the comfort of wearing a seatbelt during pregnancy, a small proportion of women reported pressure on the thorax and abdomen from the first trimester onwards, while 50% experienced abdominal pressure from mid-pregnancy and 25% reported thoracic pressure in the last trimester. These results suggest that chest and abdominal protrusion after the second trimester of pregnancy has an impact on the proper wearing of seatbelts among twin-pregnant women. The proportion of women who kept their waist belts as close to their legs as possible decreased as pregnancy progressed, as did the proportion of women who wore their seat belts during pregnancy. In the second trimester, more women did not wear shoulder belts, suggesting that removing the shoulder belt relieves pressure on the chest and abdomen. However, this method of wearing seat belts poses risks not only to the mother but also to the foetus while driving. Furthermore, in terms of seat belt use, the proportion of pregnant women who never wore a seatbelt increased to 16% by the end of the pregnancy, which is significantly higher than the 0.4% observed in singleton pregnancies [7]. This difference may be attributed to ‘abdominal pressure’, which was cited as a primary reason for not wearing a seatbelt. This indicates that physical changes unique to twin pregnancies affect behaviours regarding safety while driving. In the future, the relationship between discomfort with wearing a seatbelt and changes in the body size and shape of women pregnant with twins should be clarified at different stages of pregnancy.

In a study comparing the incidence of moderate or severe injuries in motor vehicle collisions among singleton and non-pregnant women, the incidence of abdominal injuries was higher in singleton pregnancies, although the incidence in other body regions was higher in non-pregnant women [21]. This may be due to the abdomen being more susceptible to external forces during pregnancy. Therefore, if the seatbelt is not worn appropriately, as in the results of this study, external forces may impact the abdomen through the seatbelt, or inadequate restraint may allow forward movement, leading to abdominal injuries against the vehicle’s interior structure, mainly the steering wheel. Such abdominal injuries pose significant risks to both mothers and foetuses, potentially leading to life-threatening complications, such as early placental abruption, uterine rupture, and premature water breakage. This study also revealed that the frequency of driving in twin-pregnant women decreased compared to pre-pregnancy and progressively from the first trimester to the third trimester. This decline in driving frequency suggests that twin-pregnant women experience increased abdominal load, physical load, and driving anxiety throughout pregnancy. Twin pregnancies, compared to singleton pregnancies, are associated with a higher risk of obstetric complications requiring hospitalisation and management, with morning sickness being more pronounced in the early stages and uterine enlargement being more significant in the second trimester. Given these factors, it is likely that the frequency of driving decreases in twin-pregnant women.

In light of these findings, it is imperative for midwives to inform twin-pregnant women, before or early in their pregnancy, about the potential impacts of their pregnancy on their daily lives. Educating them about proper seatbelt usage when continuing to drive can enhance their understanding and promote safer practices. It also empowers twin-pregnant women to actively consider support from their surroundings and alternative transportation options to alleviate the physical and mental stresses of pregnancy and take proactive safety measures.

In addition, discomfort due to seatbelt use in the chest and abdomen often emerges after the second trimester of pregnancy, leading several pregnant women to wear seatbelts improperly. Such improper use of seatbelts poses risks to both mothers and foetuses. Furthermore, common physical and mental changes contribute to complaints such as ‘tension and cramps in the lower abdomen’, ‘distractedness’, and ‘irritability’, which are risk factors for collisions or near-miss incidents while driving [4]. Therefore, for the safety of both mothers and foetuses, the following instructions should be given to twin-pregnant mothers at midwife outpatient clinics and motherhood classes: pregnant women should choose not to drive a car if the twin-pregnant mother is experiencing discomfort or unable to wear a seatbelt due to the discomfort caused by wearing a seatbelt. Furthermore, it is crucial that these women are made aware of the importance of consistent seatbelt use. This involves providing guidance across various aspects of women’s health, not just maternity care. Additionally, considering that pregnant women seek information from the Maternal and Child Health Handbook, acquaintances, family members, and social networking services, it is essential to ensure that they receive accurate information.

### 4.2. Proposal of a Comfortable Seatbelt for Twin-Pregnant Women

As pregnancy progresses, women pregnant with twins often experience discomfort while wearing seatbelts. Along with abdominal pressure, these women also face common pregnancy complaints, such as breathing difficulties, itchy skin, and back pain. To manage these physical changes, women pregnant with twins typically adjust their seatbelts from the first trimester to avoid abdominal protrusion, attach fabric to the belt to alleviate pressure, and position shoulder belts away from the chest. Based on previous survey responses and innovations made in seatbelt design, it has become clear that the top priority for a comfortable seatbelt for women pregnant with twins is to alleviate pressure on the chest and abdomen. In this study, 28.0% of pregnant women preferred a wider seatbelt to reduce abdominal pressure. Previous studies have reported that seatbelts that are 1.5 to 2 times wider can lessen abdominal stress during a crash and reduce the incidence of placental abruption [22]. Regarding alleviating abdominal pressure, while adjustable belts can alleviate discomfort, they may not provide sufficient restraint force or ensure safety [23]. Therefore, instead of adjusting the belt pressure, it may be more effective to use a lap belt to minimise abdominal pressure and a backpack-type shoulder belt to avoid chest pressure. The relationship between lap belt width and seatbelt pressure as perceived by pregnant women should be investigated through the use of belts of various widths, conducting objective pressure assessments at each pregnancy stage. Another approach to avoiding chest pressure might involve modifying the width of the shoulder belt beyond the backpack style. For the backpack type, considerations of how it can be integrated into the vehicle’s interior structure for safety and comfort are also necessary. The comfortable seatbelt configuration for women pregnant with twins should be validated through experiments using finite element models that simulate pregnant women with twins to ensure the safety of both mother and child.

### 4.3. Strengths and Limitations of This Study

There are several strengths of this study. First, this study compared the driving behaviours of women pregnant with twins at different stages of pregnancy, providing detailed insights into their behaviour. Second, the roles of midwives or obstetricians for women pregnant with twins were clarified. The following guidance should be given to pregnant women from early pregnancy to midway through, when abdominal protrusion is more pronounced for twin-expectant mothers. First, women should prepare their environment to assist them in their surroundings. For example, when a twin pregnancy is confirmed, human resources need to be secured, such as asking the partner or family members to take the couple to and from the hospital. Second, they should refrain from driving themselves. These guidelines will protect mothers and children from motor vehicle collisions. Third, our research has identified the essential features of an ideal seatbelt for women pregnant with twins. Pregnant women carrying twins often feel pressure in their chest and abdomen and need seat belts that alleviate this discomfort. While safety verification of this ideal seatbelt is still necessary, our study offers concrete proposals and highlights critical considerations for ensuring a comfortable pregnancy for women carrying twins. Fourth, this cross-sectional survey was conducted via an internet questionnaire. Obtaining responses longitudinally based on the number of weeks of pregnancy allowed for more detailed and accurate information to be collected in the first, second, and third trimesters.

This study has certain limitations. First, the sample size was small. However, women pregnant with twins are not common. Furthermore, low birth rates are a serious issue in developed countries including Japan. Therefore, this study provided valuable results with a limited sample size. Second, as this study was based on an online survey, we could not obtain the stature of the pregnant women. Because the interaction between the seatbelt and the body depends on the stature of the pregnant woman, a further study would clarify the interaction between the stature of women pregnant with twins and the seatbelt path.

## 5. Conclusions

This survey-based study revealed several key findings regarding automobile driving among women pregnant with twins, categorised by gestational age. As pregnancy progresses, driving frequency decreases among women who regularly drive a car. The correct seatbelt use rate is lower and the rate of never using a seatbelt is higher among women pregnant with twins than those with singleton pregnancies previously reported. To avoid pressure on the chest and abdomen, women pregnant with twins adopt specific methods of wearing seatbelts from the first trimester. However, most of them end up wearing seatbelts improperly, especially as they approach the third trimester. A comfortable seatbelt for these women would be one that can alleviate belt pressure and includes features such as a lap belt to lessen abdominal pressure, a thicker belt to distribute pressure evenly, and a shoulder belt designed similarly to a backpack. In the future, it will be crucial to provide health guidance to women pregnant with twins to help them manage their driving activities based on physical changes, evaluate their ability to drive during the third trimester, and assess the safety of seatbelts specifically designed for improving comfort.

## Figures and Tables

**Figure 1 healthcare-12-01590-f001:**
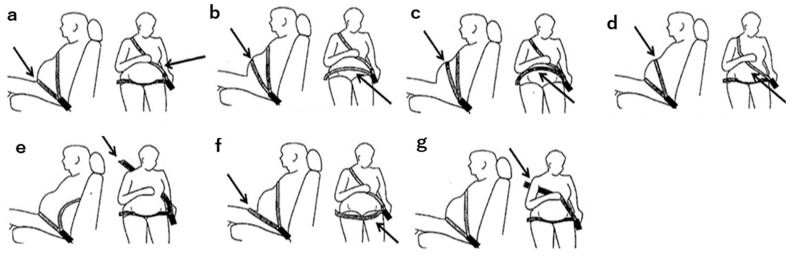
Patterns of seatbelt use. Participants indicated their seatbelt usage from seven patterns. The arrows indicate the more characteristic parts of the seatbelt wearing pattern. (**a**) Wearing a seatbelt that avoids the distended abdomen. (**b**) The lap belt crosses the distended part of the abdomen. (**c**) The lap belt crosses the upper abdomen. (**d**) The shoulder belt crosses the upper abdomen. (**e**) The shoulder belt is not worn. (**f**) The lap belt crosses the thighs. (**g**) The shoulder belt crosses the armpit.

**Figure 2 healthcare-12-01590-f002:**
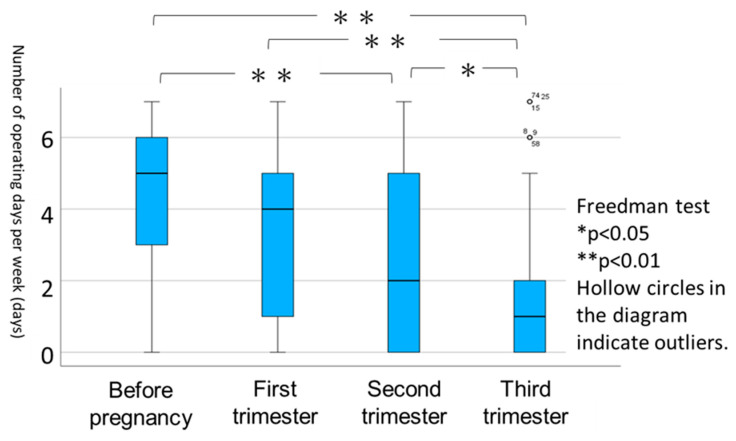
Number of operating days in a week.

**Figure 3 healthcare-12-01590-f003:**
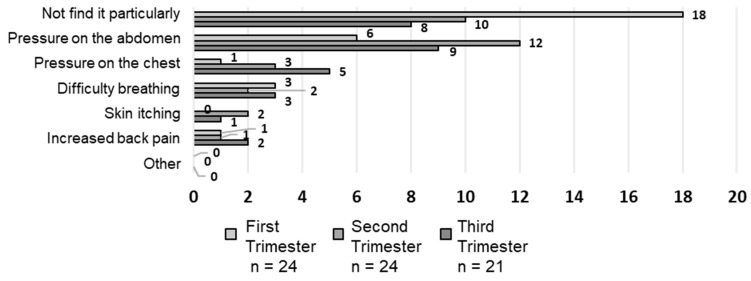
Feelings of discomfort while wearing a seatbelt (multiple responses).

**Figure 4 healthcare-12-01590-f004:**
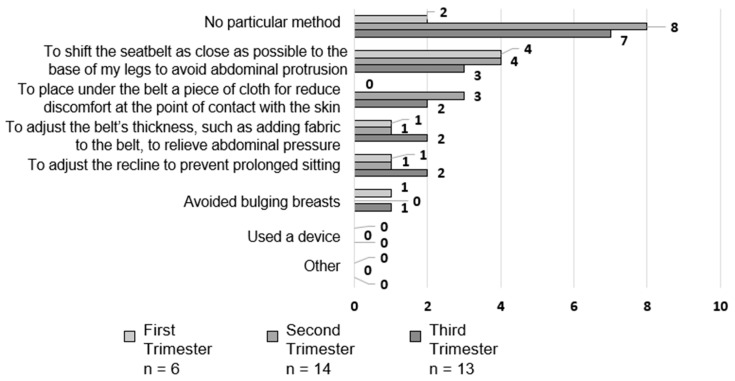
Devices used while wearing a seatbelt (multiple answers).

**Figure 5 healthcare-12-01590-f005:**
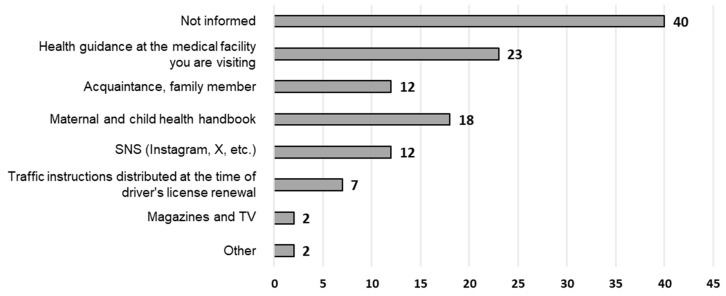
Providing information on seatbelts (multiple responses, *n* = 82).

**Table 1 healthcare-12-01590-t001:** Seatbelt use during each trimester from pre-pregnancy to the third trimester.

	Pre-Pregnancy	1st Trimester	2nd Trimester	3rd Trimester	Total (%)
Wearing a seatbelt that avoids the distended abdomen (correct method)	51 (62.2%)	13 (54.2%)	12 (50.0%)	9 (42.9%)	85 (56.3%)
Lap belt crosses the most distended part of the abdomen	14 (17.1%)	4 (16.7%)	3 (12.5%)	1 (4.8%)	22 (14.6%)
Lap belt crosses the upper abdomen	3 (3.7%)	0 (0.0%)	3 (12.5%)	0 (0.0%)	6 (4.0%)
Shoulder belt crosses the upper abdomen	4 (4.9%)	3 (12.5%)	0 (0.0%)	2 (9.5%)	9 (6.0%)
Shoulder belt is not worn	2 (2.4%) ^a^	1 (4.2%) ^a,b^	1 (4.2%) ^a,b^	4 (19.0%) ^b^	8 (5.3%)
Lap belt crosses the thighs	2 (2.4%) ^a^	2 (8.3%) ^a,b^	4 (16.7%) ^b^	2 (9.5%) ^a,b^	10 (6.6%)
Shoulder belt crosses the armpit	6 (7.3%)	1 (4.2%)	1 (4.2%)	3 (14.3%)	11 (7.3%)
Total (%)	82 (100%)	24 (100%)	24 (100%)	21 (100%)	151 (100%)

The chi-square test was performed and adjusted for multiple comparisons using the Bonferroni method. Each superscript letter indicates a significant difference (*p* < 0.05) for different letters.

## Data Availability

The original data presented in this study are openly available in Mendelery Data at 10.17632/ktgxfdwdnb.1.

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
