# Peer review of "Comfortable Seatbelts for Pregnant Women with Twins in Japan: A Cross-Sectional Observational Study on Seatbelt Usage"

_healthcare, 2024, doi:10.3390/healthcare12161590_

Round 1

Reviewer 1 Report

Comments and Suggestions for Authors

The article reports the online survey results of seatbelt-use habits  of pregnant women with twin fetuses in Japan and attempts to define 'ideal' seatbelts. It is very important to research into safety of pregnant drivers, a rather neglected population.

Once the paper is improved the authors might like to reconsider the title of the article. The numbers in the following remarks are the reference to the line numbers.

51  .. enter the workforce later in life and delay marriage..  Ref needed

54  .. with the increasing numbers of twin pregnancies .. Ref needed

66 .. more information  on contractor

67  Numbers (n=  ) are needed for each trimester group

111   Ideal Seat belt.. There is a fundamental issue here:  Researchers can identify the problems with the existing products then it is their job to use their skills and judgement to find the solution (or perfect solution) . The engineering designer might get insprired by the users idea however the user (of any product) is not necessarily an engineering designer. It is unusual to expect the user to find a solution - let alone the ideal one.  Offering possible options is also wrong on the researchers side before they know the problem. This comment  is also relevant for the title - which in my opinion must be corrected. There are far too many parameters in a seatbelt design, user comfort is only one of them.

147-153    I believe re-writing this paragraph in better English will make it more focussed

154-158    same as above

Figure 2    Where did the before pregnancy data come from? Explaination needed

201-202   When talking about statistical significance it is also important to consider the actual data and make comment on it especially when the sample is so small.

Table 1  This is the first time we see the numbers!

Figure 3 - 'Skin itching' relevance needs explanation. 

225  The percentages are intriguing - needs explanation. Or is it the other way round?

Figue 4  Use of percentages would provide better representation

242   Ideal   ?   (not really)  - Might like to use 'Opinion of .. ' 

335  ???

348  Ideal   ?  

274-399 Discussion  is mainly the repetition of the results and hypothesis;  e.g. in lines 357-358  women's desire for safety and comfort is very important  so the reserchers only need them to identify the problem as mentioned at the beginning of this feedback.

Another major weakness of this article is the comparison of habits of women with single/twin pregnancies. What is different? Why?

Finally, to improve the quality of the research and manuscript, authors should take the advantage of experience/findings resulted from extensive studies carried on  for 15 years at Loughborough University, UK on the safety of pregnant drivers. All published work (including one in this Journal). 

Reviewer 2 Report

Comments and Suggestions for Authors

The subject of your article is particularly interesting and very topical.

In order to make your article technically and scientifically robust, it will be necessary to make some improvements to the different chapters of the manuscript, signalled throughout the document.

Comments on the Quality of English Language

Round 2

Reviewer 2 Report

Comments and Suggestions for Authors

Significant improvements have been made to your manuscript. Thank you very much. 

My only suggestion would be to mention in the title that the article focuses only on Japan.

Author Response

Comments 1: Significant improvements have been made to your manuscript. Thank you very much. 

My only suggestion would be to mention in the title that the article focuses only on Japan.

Response1: Following your suggestion, we have added 'in Japan' to the title.